# Metagenomic Insights for Antimicrobial Resistance Surveillance in Soils with Different Land Uses in Brazil

**DOI:** 10.3390/antibiotics12020334

**Published:** 2023-02-05

**Authors:** João Vitor Wagner Ordine, Gabrielle Messias de Souza, Gustavo Tamasco, Stela Virgilio, Ana Flávia Tonelli Fernandes, Rafael Silva-Rocha, María-Eugenia Guazzaroni

**Affiliations:** 1Department of Biology, Faculdade de Filosofia, Ciências e Letras de Ribeirão Preto, University of São Paulo, Ribeirão Preto 14040-900, SP, Brazil; 2ByMyCell Inova Simples. Avenue Dra. Nadir Águiar, 1805-Supera Parque, Ribeirão Preto 14056-680, SP, Brazil

**Keywords:** soil, microbiota, antibiotic resistance, soil resistome, surveillance, One Health

## Abstract

Land-use conversion changes soil properties and their microbial communities, which, combined with the overuse of antibiotics in human and animal health, promotes the expansion of the soil resistome. In this context, we aimed to profile the resistome and the microbiota of soils under different land practices. We collected eight soil samples from different locations in the countryside of São Paulo (Brazil), assessed the community profiles based on 16S rRNA sequencing, and analyzed the soil metagenomes based on shotgun sequencing. We found differences in the communities’ structures and their dynamics that were correlated with land practices, such as the dominance of *Staphylococcus* and *Bacillus* genera in agriculture fields. Additionally, we surveyed the abundance and diversity of antibiotic resistance genes (ARGs) and virulence factors (VFs) across studied soils, observing a higher presence and homogeneity of the *vanRO* gene in livestock soils. Moreover, three β-lactamases were identified in orchard and urban square soils. Together, our findings reinforce the importance and urgency of AMR surveillance in the environment, especially in soils undergoing deep land-use transformations, providing an initial exploration under the One Health approach of environmental levels of resistance and profiling soil communities.

## 1. Introduction

Antimicrobial resistance (AMR), one of the most serious health risks of the 21st century, is a common competition mechanism used by environmental bacteria to ensure their survival in their natural environment. Although evidence of antimicrobial resistance in bacteria dates back to the pre-antibiotic era, studies suggest that human activity has a significant impact on the extension and diversity of the bacterial resistome [1,2,3].

Even though the soil microbiota naturally presents a large and robust diversity of ARGs in its intrinsic resistome, land-use transformation due to anthropic activities, such as the excessive use of antibiotics in livestock production [4], antibiotic-enriched manure application [5,6], and excessive use of xenobiotics in crops [7,8], along with increasing levels of deforestation for farming or urban purposes, can alter bacterial communities and disseminate ARGs throughout the environment [9]. In these highly modified sites, soil bacteria more frequently are in close proximity to commensal and pathogenic bacteria, which could lead to an increased horizontal ARG transfer rate among them, followed by a dominance of organisms with acquired resistance in comparison to intrinsically resistant bacteria [10].

The “One Health” surveillance approach takes into consideration the interrelated link among people, non-human animals, and the environment [2]. The substantial role of the latter in AMR spread can be noted in the soil’s capacity to serve as a resistance gene reservoir, facilitating the spread of ARGs found in mobile genetic elements (MGEs), such as plasmids, integrons, and transposons, among different bacterial species, speeding the development of multidrug-resistant (MDR) pathogens [11]. According to Ghosh et al. (2021) [12], MDR pathogens are thought to be responsible for up to 10 million cases of fatalities annually worldwide, with a mortality rate of 392,000 in Latin America.

Brazil, the largest country in Latin America, presents a population of 192 million people. Although the National Health Regulatory Agency (ANVISA) in Brazil has compiled data regarding healthcare-associated infections and levels of antimicrobial resistance in clinical settings over the last decades [13], the country is not equipped with a central microbiology reference laboratory, increasing the difficulty in conducting national data analysis regarding bacterial resistance [14]. The São Paulo state, located in the southeastern region in Brazil, inhabited by over 45 million people [15], is a critical region in terms of high levels of resistance among important pathogens, such as non-fermenting Gram-negative bacilli and Gram-positive cocci, such as *Staphylococcus aureus* [15].

Recent estimates suggest that Brazil was responsible for almost 8% of all antibiotic consumption for veterinary purposes globally in 2017 and has an increased consumption projection of 11.8% in 2030 [16]. This is mainly due to a shift toward intensified livestock production systems that regularly use antimicrobial agents, which can directly affect the number of ARGs disseminated through the environment and, consequently, might contribute to increased levels of AMR in clinical settings [16].

Previous studies have shown the seriousness and urgent need to tackle AMR in the countryside of São Paulo, Brazil, especially after COVID-19 pandemic [17]. During this period, high rates of antibiotic use in hospitalized patients and prolonged time in invasive therapy have caused an alarming increase in polymyxin B-resistant *Klebsiella pneumoniae* isolates in 2021 [18] and a nosocomial outbreak of extensively drug-resistant (EDR) *K. pneumoniae* in 2022 [19].

Despite the extreme importance of monitoring healthcare infections associated with antibiotic-resistant bacteria to combat AMR, there is still a lack of data regarding environmental levels of resistance in northeastern soils of the São Paulo state, where there has been historical land use of soils for urban construction, agricultural, and livestock practices, leading to a population density of approximately 1.75 million inhabitants [20,21,22]. Considering the urgent need to tackle AMR, not only in clinical settings, but also taking into account the One Health approach, we used 16S rDNA and shotgun sequencing to profile the bacterial communities and resistome of eight sites in the countryside of São Paulo, to provide the first, to our knowledge, environmental AMR surveillance of soil samples in this region. 

## 2. Results

### 2.1. Bacterial Community Dynamics

To understand the differences in bacterial communities across soils from different locations and land uses, we collected eight soils from different sites across São Paulo’s northeast region (Table 1 and Appendix A), and we assessed the community profiles based on 16S rRNA sequencing through an Oxford Nanopore MinION device, which allows full-length sequencing of the 16S gene (Figure 1A). In all sampling sites, *Bacillus* was observed as one of the most ubiquitous bacteria in the communities, with relative abundances of 1.7% in the campus lawn soils and in the PPA, 2% in the urban square, 2.4% in the orchard, 4.4% in the hen house, 11.7% in the cattle sites, 17.7% in the pasture area, and 31.1% in the agriculture soils. Two other frequent genera identified were *Vicinamibacter*, with relative abundances ranging from 1.9% (orchard) to 6.2% (pasture) and *Rhodoplanes*, ranging from 1.9% (hen house) to 4.4% (PPA soil) (Appendix A).

Additionally, in order to comprehend bacterial diversity in each area, the Shannon index was calculated for each soil sample, indicating a smaller diversity in agricultural soils (3.781), followed by campus lawn (4.343) and PPA (4.504), and the highest diversity was found in the urban square (5.056) and hen house soils (5.087), in the analyzed conditions (Appendix A). We next performed a principal component analysis (PCA) and a hierarchical clusterization (Figure 1B and Appendix A, respectively) with the resulting microbial profiles of each soil site sampled. In the former, the variance of sampling sites in two distinct groups was explained mostly by the first principal component (*PC1* = 70.8%), which created a gradient of samples according to degree of land-use change. When this result was analyzed in combination with the PC2 (13%), three apparent clusters were formed, corroborating the land-use classification used. Additionally, statistical analyses indicated significant differences (F-value = 5.229, *p*-value < 0.001) in the relative abundance of genera across sampled sites.

To further understand the dynamics of bacterial communities, we performed a correlation analysis for taxa with a minimum relative abundance of 5% (Figure 1C). As shown in the figure, *Terrimonas* and *Lysobacter* displayed the strongest positive correlation (0.92), followed by *Bacillus* and *Staphylococcus*. On the other hand, *Fimbriiglobus* and *Lysobacter* displayed the strongest negative correlation (−0.69), followed by *Fimbriiglobus* and *Pseudolabrys* (−0.64) and *Vicinamibacter* with *Occallatibacter* (−0.58).

### 2.2. Abundance and Diversity of ARGs and VFs in Soils’ Metagenomes 

Aiming to understand the abundance and diversity of antibiotic resistance genes (ARGs) and virulence factors (VFs) across studied samples, we proceeded to analyze the soil metagenomes based on shotgun sequencing through the Illumina platform (Figure 2). Using the CARD database, we detected ARGs in all sampled sites with a total of 254 ARGs identified, ranging from 17 (PPA) to 60 (cattle site) (Appendix A). The identified ARGs conferred potential resistance to eight pharmacological classes of antibiotics, with glycopeptide (77.5%), rifamycin (12.2%), and macrolide/penam (5.9%) being the most frequent ARG types across all soils analyzed, followed by trimethoprim (1.1%), phenicol (0.8%), aminoglycoside (0.4%), and isoniazid/rifamycin (0.4%) (Figure 2A).

In total, 12 ARG types were identified in all soils, with *vanRO* and *vanSO* (glycopeptide)*, rbpA* (rifamycin), and *mtrA* (macrolide/penam) being the most frequent genes, followed by *dfrB3* and *dfrB7* variants (trimethoprim), *cpt* (phenicol), *aac2-lb* (aminoglycoside), and *efpA* (isoniazid/rifamycin resistance). Additionally, three β-lactamase genes were identified, one being a serine-β-lactamase (SBL), identified as *blaF*, in urban square soils and two metallo-β-lactamase (MBL) encoding genes, identified as *bla*BJP-1 and *bla*LRA-9, in orchard soils (Figure 3A).

Our analyses indicated a higher dissimilarity between the resistome profiles of livestock soils (pasture and cattle sites) and forest soils (0.5), whereas other soils did not have such a pronounced dissimilarity in the relative abundances of ARGs (Figure 3B). Although dissimilarities were found, no statistical difference was observed in the resistome profiles across the studied sites (F-value = 1.06, *p*-value = 0.37) (Appendix A).

In order to visualize connections between the resistome (dis)-similarities of the selected soils, ARGs’ relative abundances across the studies sites were used to plot a chord diagram (Figure 4A). This indicates that even though *vanRO*, *mtrA*, and *rbpA* were widespread in all soils, a higher abundance of genes in cattle sites and pasture soils was observed. Notably, *vanRO* was the ARG with the most hits across soils, with 54 and 37 hits in cattle and pasture fields, respectively (Appendix A). Differently, a higher diversity of ARGs was identified in orchard and urban square soils, including in addition to the aforementioned β-lactamases genes, the two *dfrB* variants, *cpt*, *aac2-lb*, and *efpA* ARG types.

From the VFDB database, 143 virulence genes were detected in all soil samples, divided into 25 different genes, assigned to 10 classes referring to their virulence functions (Figure 2B). Among them, the main virulence factor was an acyl carrier protein encoded by the *acpXL* gene (Figure 4B), which corresponded to 45.5% of all VFs identified throughout the soils. Types IV and VI secretion systems together composed 22.4% of the virulence genes found (Appendix A), mostly involved in adaptation and manipulation of their environment and also in the aggravation of infectious conditions when present in pathogenic bacteria [23]. Nonetheless, genes associated with bacterial motility related to type IV pili, such as *pilT*, *pilM*, *pilG*, and *pilH*, and rotating flagella, such as *flgC*, *fliE*, *fliQ*, *fliP*, *fliN*, and *fliA* were also widespread in soils. When aligning our sequences against the PlasmidFinder database, no plasmid markers were identified in our metagenomic data.

## 3. Discussion

### 3.1. Bacterial Community Structure and Dynamics

The apparent clusterization of soils based on microbial composition in our analyses corroborated the land-use classification previously used. This could be observed, for instance, in the statistically significant difference between the agriculture field and PPA communities (adjusted *p*-value < 0.005) and the similarity between the PPA and campus lawn communities (adjusted *p*-value > 0.9). This result suggests that bacterial communities in soils are shaped and modified according to land use over the years, endorsing previous reports in the literature [24,25,26].

*Vicinamibacter* and *Rhodoplanes* genera, members of the Acidobacteriota and Pseudomonadota phyla, respectively, were ubiquitous and abundant in all soils, likely due to their essential roles in carbon, nitrogen, and sulfur cycling [27,28,29,30]. Forest soils presented smaller diversity indexes compared to urban and livestock soils (Appendix A), which corroborates a previous hypothesis that higher taxonomic diversity is essential to stressed soils maintenance [31]. The main genera found in those soils were *Lysobacter*, *Pseudolabris* and *Bradyrhizobium* (phylum Pseudomonadota), *Ocallatibacter* (phylum Acidobacteriota), and *Terrimonas* (phylum Bacteroidota) [32,33,34], with a smaller abundance of *Bacillus* species compared to other sites (<2.5%), which goes in accordance to previous studies that profiled bacterial communities in preserved soils [9,10,35].

In the other hand, urban soils presented, along with hen house soils, the highest diversity indexes (Appendix A), which could be explained by the accumulation of human activity wastes and by-products in the former, as well as the introduction of gastrointestinal microbiota members in the latter [36]. Both soils were abundant with *Massilia* (phylum Pseudomonadota), common environmental bacteria that have been shown to cause opportunistic infections in immunocompromised patients [37,38]. Farming soils presented a higher abundance of *Bacillus*, especially in livestock and agricultural soils (relative abundances of 12–31%), indicating a dominance of this group in farming land-use systems in São Paulo’s northeastern soils, corroborating previous reports [39,40]. This could be explained by a spore-forming characteristic of *Bacillus*, which facilitates their high resistance to most adverse environmental conditions on farming land-use systems, such as heat, desiccation, and high levels of UV radiation [33,41,42].

In agricultural soils, *Staphylococcus* and *Bacillus* represented the majority of sequenced members (Figure 1A), which could be connected to the 84% positive correlation observed (Figure 1C). In addition, the microbial diversity in agriculture fields was smaller compared to other soils, which could be attributed to soil microbial community homogenization due to the intensified land use of sugarcane crop soils [43,44], such as the one collected for this study. It is important to note that some environmental microorganisms, such as *Staphylococcus* and *Bacillus* identified in the studied soils, not only are commonly found in the environment due to their important interactions with other bacteria and functional maintenance in soils, but can also infect humans and other animals [45]. An imbalance caused by anthropic activities on soil microbial communities could favor *Staphylococcus* and *Bacillus* species due to their early proliferation characteristics [46], taking advantage of transient conditions to outgrow more fastidious microorganisms. Thus, the early proliferation of these bacteria, along with positive interactions between them, could explain their dominance in agricultural fields [47,48].

Members of the genus *Bacillus* are among the most abundant bacterial genera found in soils, with a widespread distribution through different ecological niches, which goes in accordance to our findings [49]. Although *Bacillus* species were sequenced in all studied soils, a higher prevalence of the genus was observed in farming soils, which corroborates previous reports [50,51]. In spite of the fact that the majority of *Bacillus* species are strictly environmental, common mechanisms used for environmental competition and cell survival can aid the infection process of vertebrate hosts in certain strains, allowing these bacteria to occupy an additional niche [52,53]. *Bacillus anthracis*, for instance, can persist for many decades in soils as endospores and, when inhaled by humans or grazer livestock animals, can result in the anthrax disease [54,55]. Nonetheless, human exposure to *B. anthracis* spores has been associated over the years with agricultural contact or along cattle trails, both in South and in North America [56,57].

On the other hand, *Staphylococcus* spp. Are common in the biosphere and possess the capacity to withstand extreme temperatures and pH variations, allowing them to occupy different niches, including soil, water, non-human animals, and humans [58,59]. Comparably, other studies that profiled soil microbiota have also shown a high abundance of this genus in the microbial community [60,61]. In this sense, *Staphylococcus aureus* is one of the most prevalent bacteria in the genus, being part of the microbiota as a commensal organism or the agent of several diseases, such as dermatitis and urinary, gastrointestinal, and respiratory tract infections [62,63]. Leung and collaborators have correlated a higher number of *S. aureus* in the environment with the severity and persistence of atopic dermatitis in the United States [64]. In that sense, the substantial abundance of *Staphylococcus* in agriculture fields (5.8%) could suggest a concerning scenario, due to the high circulation of people in the area and the broad range of niches the bacterium can occupy, reinforcing the importance of more studies with a One Health approach.

### 3.2. Resistance Genes and Virulence Factors Identification

Although no statistical difference was observed among the resistome profiles of soils with different land uses, dissimilarities were identified (Figure 3B). One of the most pronounced results was the notorious number of genes related to glycopeptide resistance (*vanRO* and *vanSO*) in livestock soils, totaling 91 ARG hits within these soils (Appendix A). Differently, forest soils presented a smaller number of hits for the same gene (28 hits), with a threefold difference compared to the total hits in livestock soils. It is also worth mentioning that *vanRO* was identified in all sampled soils and *vanSO* in 38% of them (Figure 3A). Both *vanRO* and *vanSO*, components of the same *vanO* gene cluster that can potentially confer glycopeptide resistance, were first identified in *Rhodococcus equi* soil isolates in Denmark [65]. Glycopeptide antibiotics, such as vancomycin, are a last-resort treatment option for methicillin-resistant *S. aureus* (MRSA) and vancomycin-resistant enterococci (VRE) infections [66,67]. In Brazil, several waves of resistance of *S. aureus* against antimicrobials have been reported, with increasing numbers of MRSA strains isolated in different hospitals in São Paulo, one of the most affected in recent decades [61,66,68]. Additionally, glycopeptides, such as avoparcin, have been historically used as growth promoters in the livestock industry, with worldwide reports of VRE in cattle, poultry, and swine samples [69,70]. As a result of this, glycopeptide-resistant genes are commonly reported as abundant in fecal samples [71,72], which could explain the higher number of *vanRO* hits in livestock and poultry soils in our study. Nonetheless, other environmental studies have reported the presence of these genes in permafrost samples from over 10,000 years ago and throughout the environment, suggesting an innate resistance reservoir in the soil microbiome [73,74]. Although vancomycin resistance genes are commonly found in soils worldwide [74], our findings highlight the importance of environmental surveillance, given that the *Staphylococcus* genus was one of the major bacteria present in agricultural soils, and this soil showed a high abundance of *vanRO* genes. This, added to the facilitated route of transmission of *vanRO* from livestock to humans, either through direct contact or by the food chain, reinforces the need for monitoring of these soils [70].

Macrolide antibiotics act by binding to the bacterial 50S ribosomal subunit, causing the cessation of bacterial protein synthesis, as a bacteriostatic agent [75]. The broad antibacterial activity of this antimicrobial has led to its widespread use in gastrointestinal and respiratory tracts, and in sexually transmitted infections [76]. In staphylococcal infections, there is an increasing cross-resistance to macrolides in MRSA strains, categorizing these bacteria as pathogens of great concern [77]. β-lactam antibiotics share the presence of a β-lactam ring in their structures, with a broad-spectrum activity due to the penicillin-binding protein (PBP) inactivation that hampers cell wall formation [77]. These are the most prescribed antibiotic classes in clinical settings worldwide, with annual expenses of approximately US $15 billion, representing 65% of the total antibiotic market [78,79]. In the last few years, the dissemination of Gram-negative bacteria resistant to β-lactams has been considered a public health threat, especially when considering the absence of new antibiotics with activity against these bacteria in the last 20 years [80]. The transcriptional activator of the *mtrCDE* multidrug efflux pump, *mtrA* (widespread through all sampled soils of this study), is responsible for expressing the operon that exports a wide variety of antimicrobial agents, including β-lactams and macrolides [81,82,83]. The aforementioned gene has been previously reported as abundant in soils, especially in those that have undergone land-use conversion [83,84,85].

Rifamycin resistance genes have been reported as abundant in both pristine and highly modified soils, which goes in accordance with our findings [86,87], although, to the best to our knowledge, no previous studies have reported such a robust presence of the *rbpA* gene in farming soils. Its genetic product is an RNA-binding protein that is responsible for conferring low resistance levels in the soil bacterium *Streptomyces coelicolor* [88]. In addition, Bortoluzzi and collaborators have pointed out that this gene could account for the transcriptional activity in *Mycobacterium tuberculosis* against rifamycin antibiotics [89]. In 2019, 73,000 new cases of tuberculosis (TB) and 4500 deaths due to this disease were reported, with several of them related to rifampicin-resistant strains [89,90]. Additionally, genomic characterization of the zoonotic and human-opportunistic pathogens *R. equi* and *Mycolicibacterium peregrinum* obtained from human, pig, and soil samples in Asia indicated the presence of the *rbpA* gene in all isolated genomes [91,92,93]. Nonetheless, the authors of the aforesaid study suggest that infections caused by these antibiotic-resistant bacteria might have an environmental source [93]. Moreover, in the farming soils of the present study (*rbpA*-enriched), the genera *Rhodococcus*, *Mycolicibacterium*, and *Mycobacterium* were identified as minor components of the soil microbiota, which represents a cause for concern. To the best of our knowledge, there have been no reports of the *rbpA* gene in Brazilian clinical settings, but the presence of the ARG in soils containing opportunistic pathogens or in close proximity to humans and livestock could pose a threat if the gene is transferred through the food chain or to pathogenic bacteria. Thus, studies under a One Health approach are of extreme importance when it comes to understanding possible environmental sources of ARGs and which opportunistic pathogens are present in the environment.

While fewer genes represent the majority of ARGs in livestock soils, the highest microbial and ARG diversity was identified in urban square and orchard soils, including β-lactamase-encoding genes. The products of these genes might confer resistance to most of the drugs included in the β-lactam class, which correspond to the vast majority of less toxic options used to treat bacterial infections [78,79]. These enzymes are capable of inactivating β-lactam antibiotics and can be classified either as SBLs, with an active site containing a catalytic serine residue [93], or as MBLs, which use zinc as a cofactor for catalyzation [94]. The two MBL-encoding genes (*bla*LRA-9 and *bla*BJP-1) identified in orchard soils are categorized in the B3 MBL subclass and were previously reported in environmental samples in China, Japan, and Alaska [95,96], conferring high levels of resistance when expressed in *Escherichia coli* clones [97,98].

Although no reports, to our knowledge, of the aforementioned MBLs have been performed in clinical settings, the occurrence of these genes in orchard soils could pose a threat to human health if they migrate to pathogens. For example, blaBJP-1 confers less sensibility to chelating agents compared to other MBLs and a high catalytic activity with meropenem—a watch group antibiotic [99]. Thus, the transfer of these ARGs could lead to a risk of selection of bacterial resistance that should be prioritized as targets of stewardship programs and monitoring [99]. Carbapenem antibiotics have a broad activity spectrum against the majority of pathogenic bacteria, thus their classification as a “last-resort” treatment option [100]. These antibiotics show strong performance against extended-spectrum β-lactamases, but may be more susceptible to MBLs [101,102]. Although intrinsic carbapenem resistance is presented by some bacterial species due to the production of endogenous MBLs, acquired resistance, caused by horizontal gene transfer, is more common in clinically important bacteria, which highlights the potential thread related to the presence of MBLs in the studied soils [103].

On the other hand, the SBL-coding gene identified in urban square soils, *blaF*, is a chromosomally encoded class A β-lactamase [104]. This ARG has been previously reported in China and Rio de Janeiro (Brazil), usually identified in nontuberculous mycobacteria, organisms commonly found in soils and water, also causing opportunistic infections in humans [105,106]. It has shown broad-spectrum activity against most β-lactam antibiotics, with the exception of third-generation cephalosporins [107,108]. Few studies have indicated the presence of this β-lactamase in soils, reinforcing the need for environmental surveillance in highly modified soils, aiming to provide further information regarding environmental β-lactamases and their associated risks to human health.

Considering the results related to virulence factor identification in the eight metagenomes, the *acpXL* gene was found in higher prevalence and it encoded an acyl carrier protein, required during the process of adding very long-chain fatty acid (VLCFA) to lipid A [109,110]. LPSs are known for their role in bacterial invasion, an essential function for the host infection process, and in bacterial adaptation in the environment, regardless of established mutualistic or pathogenic interactions [110]. The VLCFA attached to lipid A has been found in most members of the Rhizobiaceae family, as well as in the *Bradyrhizobium* genus, both found in the soils of our study. VLCFA presence could confer greater tolerance to stress and adaptation in distinct habitats [111,112] due to the stability conferred to the external membrane. In addition to its functions against stress, VLCFA can also be found in pathogenic or intracellular strains, such as the pathogen *Brucella*, for example, in which the linked lipid A ensures poor recognition by innate immunity [112].

Protein secretion systems, the second most common virulence class found in our study, are used for bacterial cells to interface with their environment through interaction and manipulation, where the secreted proteins can act as virulence factors, allowing these interactions [113,114]. The type VI secretion system (T6SS), with relative abundance of 11.2% in our samples (Figure 2B), is established by the VipA protein. This complex acts as a specialized bacterial nanomachinery that releases protein particles to other cells or to the environment, allowing bacteria to interact with their surrounding environment [115,116]. Thus, it consequently acts as an important determinant of the pathogenicity of eukaryotic cells, as well as in their competitive fitness in the community [116,117]. This indicates that secretion systems have a key role in shaping the microbiota of many ecological niches and explains the ubiquity of the T6SS-related genes across soils [117,118].

On the other hand, the second VF with the highest relative abundance found in our study was the secretion system type VII (T7SS). This system is a specialized protein secretion machinery that transports substrates through the cell envelope, widespread in Gram-positive members of the Actinomycetota and Bacillota phyla, abundant across all studied soils [119,120,121]. Finally, it is important to highlight that the overall abundance of the VF could be related to important housekeeping functions of T7SS, such as sporulation, conjugation, and cell wall stability, given that it is widespread among pathogenic and environmental microorganisms [122].

## 4. Materials and Methods

### 4.1. Study Area and Sample Collection 

The samples used in this study were collected from different sites across São Paulo’s northeast region (Appendix A and Table 1). Approximately 50 g of samples were aseptically taken from the upper 10 cm layer, after a 5 cm removal of litterfall, and placed in sterile Falcon tubes. For each selected site, 3 samples were collected and mixed for a better representation of the microbial community within. In total, eight samples were collected from: (i) a permanent preservation area (PPA) in the University of São Paulo campus—Ribeirão Preto, São Paulo; (ii) the University of São Paulo campus lawn—Ribeirão Preto, São Paulo; (iii) an agriculture field—Sertãozinho, São Paulo; (iv) a pasture field—São Carlos, São Paulo; (v) a livestock site—Taquaritinga, São Paulo; (vi) a hen house—São Carlos, São Paulo; (vii) an orchard field—São Carlos, São Paulo; and (viii) Urban square—Sertãozinho, São Paulo.

All soils were allocated in three categories, according to their land uses over the last decades, with those being (i) farming—sugarcane field, livestock site, hen house, pasture field, and orchard; (ii) urban—urban square; and (iii) forest—PPA and campus lawn.

### 4.2. DNA Extraction and Sequencing

The metagenomic DNA of each soil sample was extracted using DNAeasy Powersoil® Kit (QIAGEN), following manufacturer’s recommendations. The quantification and quality analysis of the extracted DNA was performed using Nanodrop™ One (Thermo Fisher Scientific) and by an agarose (1%) electrophoresis gel. Part of the extracted DNA was used for triplicate amplification of 16S rDNA, following Nanopore—16S Barcoding Kit 1-24 (SQK-16S024, Oxford Nanopore Technologies—ONT) recommendations, adding different barcodes for each replicate. All samples were individually purified, quantified by Nanodrop™ One, and mixed in proportioned amounts in order to make a representative pool of all soil samples. A single multiplex sequencing was performed using the aforementioned kit and Flongle flowcells in MinION model Mk1B. The remaining extracted DNA was submitted to metagenomic sequencing on an Illumina NovaSeq 6000 platform at Novogene (Sacramento, CA, USA), with a sequencing depth of 12 Gb/sample. Appendix A shows the data quality summary of raw data from shotgun sequencing.

### 4.3. Data Processing and Analysis

#### 4.3.1. Amplicon Sequencing

Processing and analysis of the 16S rDNA reads were performed as recommended by de Siqueira and collaborators (2021) [123]. Briefly, reads were base-called using Guppy Base-calling Software (version 6.1.3) with the dna_r9.4.1_450bps_hac.cfg configuration file [124]. Base-called reads had their quality assessed by NanoStat (version 1.6) and NanoFilt (version 2.8) was used to select reads with quality scores above Q7 [125]. After the initial filtering step, demultiplexing of reads was performed by Porechop (version 2.4) using the barcodes from 16S Barcoding Kit 1-24 (SQK-16S024). Demultiplexed reads were mapped to a 16S rDNA NCBI reference database using minimap2 (version 2.17) [126].

#### 4.3.2. Shotgun Sequencing

Shotgun sequencing raw data were processed with the fastp (version 23.1) tool (https://github.com/OpenGene/fastp, accessed on 17 November 2022) for adapter and low-quality reads removal [127]. High-quality reads assembly was carried out with the MegaHIT tool (https://github.com/voutcn/megahit, accessed on 17 November 2022) and the metagenome annotation was performed with Prokka (https://github.com/tseemann/prokka, accessed on 17 November 2022) [128], with the assembly statistics calculated with assembly_stats (https://github.com/sanger-pathogens/assembly-stats, accessed on 17 November 2022). The identification of ARGs, virulence factors (VFs), and plasmid markers was performed with the ABRICATE pipeline (https://github.com/tseemann/abricate, accessed on 17 November 2022), with an identity cut-off of 80%, by searching previously annotated genes in reference databases (ARG-ANNOT, CARD, PlasmidFinder, ResFinder and VFDB) [129,130,131,132,133].

### 4.4. Statistics and Graphical Representation

All statistical analyses were performed using R version 4.1.0. Differences between microbiota composition in the studied soils were measured using one-way ANOVA with Tukey post hoc test for determination of significance levels. A comparison was considered statistically significant at an adjusted *p* value < 0.05. Differences between resistome profiles in the studied soils were calculated using PERMANOVA. Vegan (version 2.5.7) [134] and Tidyverse (version 1.3.0) [135] packages were used for data manipulation and processing. Graphical representation was performed using ggplot2 [136] and Flourish Studio (https://flourish.studio, accessed on 22 November 2022).

## 5. Conclusions

The concept of One Health highlights that human health is interconnected to the health of other members on ecosystems, such as soils, animals, and plants. In that sense, microorganisms are crucial in One Health, given that they are the links among all these members, seen in the role of commensal bacteria in driving the organisms’ fitness, as well as maintaining key soil functions. Here, we showed that structure and composition of the microbial communities of soil samples correlate to its land use, along with concerning interactions among environmental and opportunistic pathogens in soils that have undergone land conversion. Our study has also shown several commonly found ARGs in soils that are also responsible for antibiotic resistance in potential pathogenic bacteria that are widespread in soils, with the highest abundances present in livestock soils, providing the first, to our knowledge, environmental AMR surveillance of soil samples in a crucial region of Brazil, in terms of population density and economic relevance. Although ARGs found in the soil samples in this study may confer resistance against competitors in these habitats, their gene products may also serve other functions in soils. Thus, we reinforce that ARG or VF hits within samples do not indicate actual antibiotic resistance or actual virulence determinants. Nevertheless, identifying β-lactamases in highly modified soils highlights the importance of environmental surveillance to pull the brakes and gather more information regarding resistance levels in regions at risk for higher selective pressure due to anthropic activities.

## Figures and Tables

**Figure 1 antibiotics-12-00334-f001:**
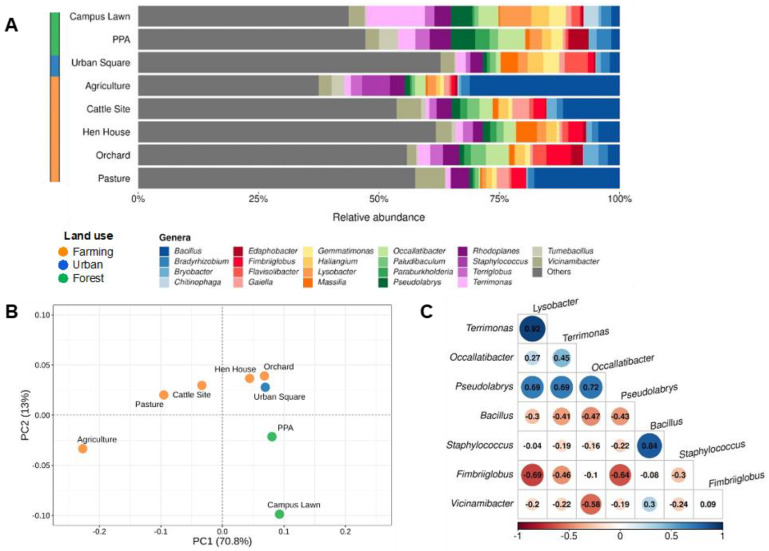
Most abundant taxa and their distributions throughout the study sites at genera level. (**A**) Relative abundances of groups found in environmental samples according to 16S sequencing. All taxa found with relative abundances below 2.5% in each sample were labeled as “Others”. (**B**) Principal component analysis (PCA) of most abundant bacteria, clustered together per sampling site and colored according to land-use classification. (**C**) Correlogram indicating negative (in red) and positive (in blue) correlations among most abundant taxa (minimum relative abundance of 5%) in the studied sites.

**Figure 2 antibiotics-12-00334-f002:**
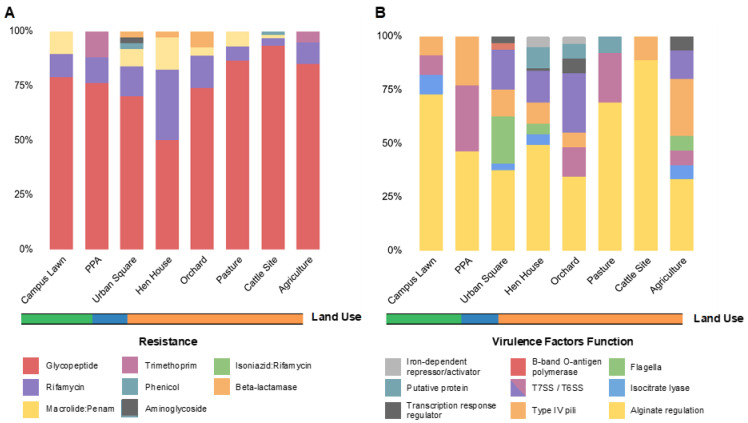
Relative abundances of ARGs and VFs per soil metagenome. (**A**) Relative abundances of the identified ARGs per soil metagenome, colored by resistance category, as indicated in the legend. (**B**) Relative abundances of the identified VFs per soil metagenome, colored by function, as indicated in the legend. The ARGs were identified by CARD and VFs by VFDB, and the abundances were estimated by dividing the number of distinct resistance genes in the category (i.e., ARG potential resistance or VF function) by the total number of genes for all classes found in that site. Land use is indicated in orange (farming), blue (urban) and green (forest).

**Figure 3 antibiotics-12-00334-f003:**
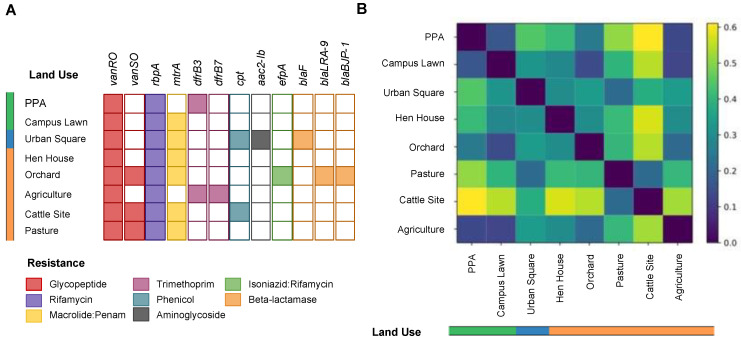
Resistome profiles across studied soils. (**A**) Distribution of different ARGs per soil metagenome, colored by resistance category, as indicated in the legend. The absence of color indicates the absence of occurrence of the ARG in the soil. (**B**) Heatmap showing the (dis-)similarity between two soil resistome profiles using Bray–Curtis distance. The purple to yellow scale (0–0.6) indicates the degree of dissimilarity between the ARGs’ relative abundances in two soils, as indicated in the legend. Land use is indicated in orange (farming), blue (urban) and green (forest).

**Figure 4 antibiotics-12-00334-f004:**
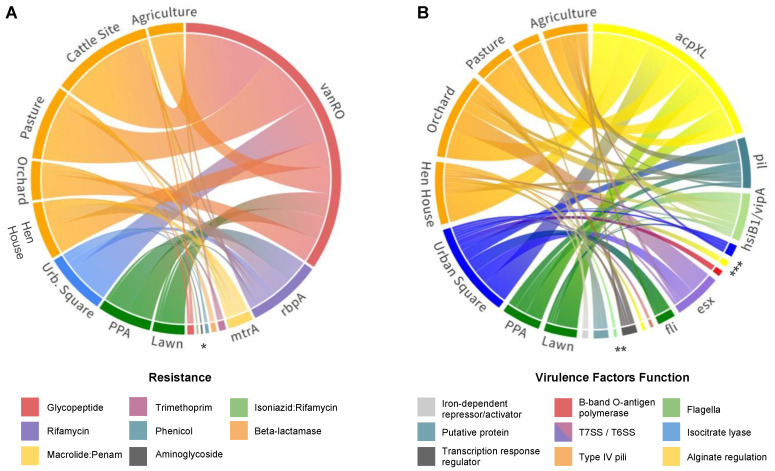
Identified gene distributions across sequenced soil samples. (**A**) Chord diagram showing the distribution of different ARGs per soil metagenome. Resistance genes are colored by resistance category and soil sites are colored by land-use system as follows: orange—farming; blue—urban; green—forest. (**B**) Chord diagram showing the distribution of different VFs per soil metagenome. Virulence-related genes are colored by function, as indicated in the legend. (*acpXL*, *alg*, *mucD*) Acyl carrier proteins; (*icl*) isocitrate lyase; (*hsiB1/vipA*) type VI secretion system, T6SS; (*flg, fli, cheW*) Flagella; (*pil*) type IV pili; (*esx*) type VII secretion system, T7SS; (*waaG*) B-band O-antigen polymerase; (*phoP*) possible two-component system response transcriptional positive regulator; (*mbtH*) putative protein; (*ideR*) iron-dependent repressor and activator. * (*vanSO, efpA, aac2-lb, cpt, bla, dfrB*); ** (*ideR, mbtH, phoP, mucD, waaG*); *** (*flgC, alg, icl*). Resource: https://public.flourish.studio/visualisation/116771 & https://public.flourish.studio/visualisation/116970. Accessed on 22 November 2022.

**Table 1 antibiotics-12-00334-t001:** Main characteristics and soil sample locations.

Soil Samples	Geological Classification	Land Use Classification	Anthropic Activity	Sampling Site	Coordinates
Agriculture (Agri)	Red Latosol	Farming	Large sugarcane crop site + Agrochemical use	Sertãozinho, SP	−21.16643, −47.99004
Pasture (Pas)	Deep Quartz Sand	Farming	Small familiar cattle farm	São Carlos, SP	−21.921, −47.90373
Cattle Site (Catt)	Bauru Sandstone	Farming	Large cattle site. High circulation of people and livestock	Taquaritinga, SP	−21.48066, −48.54118
Orchard (Orch)	Deep Quartz Sand	Farming	Small familiar orchard site	São Carlos, SP	−21.92674, −47.87008
Hen House (Hen)	Deep Quartz Sand	Farming	Small familiar poultry farm	São Carlos, SP	−21.95133, −47.89079
Urban Square (UrbSq)	Red Latosol	Urban	High circulation of people and small animals	Sertãozinho, SP	−21.1134, −47.98762
USP * Permanent Protection Area (PPA)	Red Latosol	Forest	Protected area. Access restricted to research	Ribeirão Preto, SP (USP)	−21.1662, −47.86036
USP Campus Lawn (Lawn)	Red Latosol	Forest	High circulation of people and animals	Ribeirão Preto, SP (USP)	−21.16511, −47.85944

* University of São Paulo (USP).

## Data Availability

Soil metagenomes were deposited in the SRA repository under the BioProject number PRJNA900430 (Appendix A).

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
