# Peer review of "Metagenomic Insights for Antimicrobial Resistance Surveillance in Soils with Different Land Uses in Brazil"

_antibiotics, 2023, doi:10.3390/antibiotics12020334_

Round 1
Reviewer 1 Report
The manuscript regards the significant issue, which is the problem of spreading antibiotic resistance in the environment, in this case in the soil. The study poses a part of the concept of „One Health” and I find the research idea interesting. However, several aspects need to be improved.
1. First of all, I suggest the authors consider adding the term "preliminary study" to the title of the paper. The idea and implementation of the research are interesting, but I believe that the scale of the research and the conclusions drawn are an introduction to the analysis of the problem. Moreover, the authors selected 8 sampling sites and divided them into 3 categories, with two sites in the "forest" category and only one in the "urban" category. I believe that this is not enough to be able to draw strong conclusions regarding the relationship between the type of land use and the analyzed factors.
2. In Figure 1, it is worth describing the PC1 value (>70%) and emphasizing that this axis distinguished two clusters of sampling sites, different from each other.
3. A huge gap in the whole study is the lack of statistical analyses! Authors should perform statistical analyzes for the manuscript to have real value.
4. One of my objections to the manuscript, especially since it is a microbiological study, is the omission or incorrect naming of taxonomic levels of microorganisms. Examples:
Lines 147: „In all sampling sites, Bacillus was observed...” - it should be clarified that this is a genus
Line 258: „The main groups found in those soils were Pseudomonadota (Lysobacter, Pseudolabris and Bradyrhizobium), Acidobacteriota (Ocallatibacter) and Bacteroidota (Terrimonas)…” - it should be clarified that the authors write about phylum and genus. The word "groups" in this sentence is not correct.
Please review the entire manuscript and make corrections
5. In figure 3A, I suggest explaining that the absence of color is the absence of occurrence if I understand correctly that this is what the authors meant.
6. I believe that the whole of Chapter 4 is too poor in terms of discussing the results, and too much theory in it. I suggest that you look for the right articles and emphasize how the described differences in the structure of the microbiota and the presence of ARGs fit into the threat of spreading antibiotic resistance and what are the routes of possible transmission. Moreover, I think the results should be additionally discussed with other literature, not only territorially but also taking into account the results obtained by other researchers. I also recommend emphasizing the importance of the entire work more expressively.
Reviewer 2 Report
This manuscript describes interesting research on the distribution of microbial species, antimicrobial resistance genes and virulence factors in different types of soil near Sao Paolo, Brazil.
The presentation of the data and the discussion can be considerably improved and the use of the English language is not optimal either.
Figure 1B is unclear. PCA can be done in several ways, on different parameters. What exactly is presented is not explained in the legend, nor in the short piece of text devoted to it (lines 158-60). Figure 4 is hampered by the change of the color of the lines. It would be more clear if the color of the origin was maintained.
The discussion is more a review than a proper discussion of the dataset presented in the results section. First of all the language is more complex than necessary, with too many long and convoluted sentences. In addition, it meanders between many unconnected thoughts and finally it is so long that the reader gets lost completely. I advise the authors to discuss their data in the framework of the exiting literature and focus on those aspects that combined support their conclusion.
Round 2
Reviewer 1 Report
I still think that the manuscript title should include the term "preliminary case study". Nevertheless, the authors revised the manuscript and now it presents a scientific value.
Reviewer 2 Report
The authors did a thorough revision and I am satisfied that the manuscript can be published after some minor editorial checks.